# Estimating the Spatial Accessibility to Blood Group and Rhesus Type Point-of-Care Testing for Maternal Healthcare in Ghana

**DOI:** 10.3390/diagnostics9040175

**Published:** 2019-11-05

**Authors:** Desmond Kuupiel, Kwame M. Adu, Vitalis Bawontuo, Duncan A. Adogboba, Tivani P. Mashamba-Thompson

**Affiliations:** 1Discipline of Public Health Medicine, School 0.0of Nursing and Public Health, University of KwaZulu-Natal, Durban 4001, South Africa; Mashamba-Thompson@ukzn.ac.za; 2Research for Sustainable Development Consult, Sunyani, Ghana; bawontuovitalis@yahoo.com; 3Adu Manu Kwame Consult, Accra, Ghana; meous007@gmail.com; 4Faculty of Health and Allied Sciences, Catholic University College of Ghana, Fiapre, Sunyani, Ghana; 5Regional Health Directorate, Ghana Health Service, Upper East Region, Bolgatanga, Ghana; alemna@gmail.com

**Keywords:** spatial accessibility, blood group, rhesus type, point-of-care testing, maternal healthcare, Upper East Region, Ghana

## Abstract

Background: In Ghana, a blood group and rhesus type test is one of the essential recommended screening tests for women during antenatal care since blood transfusion is a key intervention for haemorrhage. We estimated the spatial accessibility to health facilities for blood group and type point-of-care (POC) testing in the Upper East Region (UER), Ghana. Methods: We assembled the attributes and spatial data of hospitals, clinics, and medical laboratories providing blood group and rhesus type POC testing in the UER. We also obtained the spatial data of all the 131 towns, and 94 health centres and community-based health planning and services (CHPS) compounds providing maternal healthcare in the region. We further obtained the topographical data of the region, and travel time estimated using an assumed tricycle speed of 20 km/h. We employed ArcGIS 10.5 to estimate the distance and travel time and locations with poor spatial access identified for priority improvement. Findings: In all, blood group and rhesus type POC testing was available in 18 health facilities comprising eight public hospitals and six health centres, one private hospital, and three medical laboratories used as referral points by neighbouring health centres and CHPS compounds without the service. Of the 94 health centres and CHPS compounds, 51.1% (48/94) and 66.4% (87/131) of the towns were within a 10 km range to a facility providing blood group and rhesus type testing service. The estimated mean distance to a health facility for blood group and rhesus POC testing was 8.9 ± 4.1 km, whilst the mean travel time was 17.8 ± 8.3 min. Builsa South district recorded the longest mean distance (25.6 ± 7.4 km), whilst Bongo district recorded the shortest (3.1 ± 1.9 km). The spatial autocorrelation results showed the health facilities providing blood group and rhesus type POC testing were randomly distributed in the region (Moran Index = 0.29; z-score = 1.37; *p* = 0.17). Conclusion: This study enabled the identification of district variations in spatial accessibility to blood group and rhesus type POC testing in the region for policy decisions. We urge the health authorities in Ghana to evaluate and implement recommended POC tests such as slide agglutination tests for blood group and rhesus type testing in resource-limited settings.

## 1. Introduction

Since 1990, the world has made significant progress in reducing maternal mortality [1]. Despite the progress made, recent evidence shows that every eleven seconds a pregnant woman dies somewhere in the world according to the World Health Organisation (WHO) [1]. In 2017, approximately 295,000 women died from mostly preventable causes during and subsequent to pregnancy and childbirth; 94% of these mortalities occurred in resource-limited settings [2]. Sub-Saharan Africa (SSA) and Southern Asia alone accounted for 86% of all maternal deaths in the world [2]. The WHO estimates show that in 2017, SSA countries including Ghana alone accounted for almost two-thirds (196,000) of maternal deaths compared to Southern Asia which accounted for approximately one-fifth (58,000) [2]. Haemorrhage remains a major direct cause of maternal death and together with hypertensive disorders and sepsis accounts for more than 50% maternal deaths globally [3]. A recent global systematic review by Say and colleagues revealed that haemorrhage accounted for about 27.1% ahead of hypertensive disorders with 14%, and sepsis (10.7%) of the total 60,799 maternal deaths retrieved from 23 eligible studies published from 2003 to 2012 [3]. Blood transfusion service is one of the critical interventions for the management of haemorrhage [4].

Blood transfusion saves lives and improves health, but many patients, including pregnant women or women during delivery, requiring transfusion do not have timely access to safe blood [5]. A blood transfusion may become crucial at any time in both urban and rural communities [5]. Improving access to safe blood transfusion relies partly on the ability of the health facility to effectively perform point-of-care (POC) testing for blood group and rhesus type. Generally, access to healthcare services including blood group and rhesus type POC testing may also be influenced by several factors such as the availability of laboratory services, human resource capacity, availability of POC tests and supply chain management of the diagnostic tests, cost, wealth, quality of care, occupation, cultural practices, education, and the location of the service [6,7,8,9]. The nature of the road and networks, type of transport systems, topology, land use, building use, traffic condition and population density, and seasonal variations may also influence travel time to healthcare facilities, particularly in rural areas [7,10]. In some settings, these factors may interact in a complex way and geographical access in terms of distance and travel time may be insignificant [7]. Nonetheless, where the availability of POC testing for blood group and rhesus type is poor and laboratory services are sparse, the geographical location of the service becomes a key barrier to the service, particularly for the rural populations and, hence, very essential [7,11,12,13].

In Ghana, evidence shows that the maternal mortality ratio presently stands at 319 per 100,000 live births [1,13,14,15]. Haemorrhage is one of the direct causes of maternal deaths in Ghana and among the top five causes [16,17]. The Der et al. study in 2013 identified haemorrhage as the topmost cause of maternal mortality in Ghana accounting for 21.8% of all deaths with abortion, hypertensive disorders, infections, and ectopic gestation accounting for 20.7%, 19.4%, 9.1%, and 8.7%, respectively [17]. Many interventions including the implementation of a free maternal healthcare policy since 2003, provision of emergency obstetric care, expansions of health infrastructure, increased training of midwives and posting them to underserved communities, and ongoing investment in primary healthcare (PHC) facilities such as health centres and community-based health planning and service (CHPS) facilities are meant to reduce maternal deaths in the country. Pregnant women in Ghana are also expected to undertake a blood type screening test as part of the wide range of healthcare services rendered to them during the first antenatal care (ANC) visit irrespective of the level of care [18,19]. Like most SSA countries, Ghana’s PHC facilities often do not have laboratories to perform blood type screening tests for patients and donors as well as facilities for safe storage of blood. In 2018, a cross-sectional study was conducted aimed to investigate the availability and use of pregnancy-related point-of-care (POC) tests in Ghana’s PHC clinics [20]. Of the 100 participating PHC clinics in the survey, blood group and rhesus type testing was available in only six clinics with some form of laboratory services in the Upper East Region (UER) [20]. Whilst the average ANC clinic attendance per month was shown to be 65 pregnant women with a minimum of 30 and a maximum of 360 [20]. The results of the survey also revealed that out of the 94 clinics without blood group and rhesus type testing, 89 demonstrated the need for it in their clinics [20]. The findings of the survey revealed blood group and rhesus type testing is still a laboratory-based test in Ghana performed by trained laboratory professionals and, hence, may not be accessible to all who need it [20].

Despite this, to date, no study has measured the spatial accessibility in terms of distance and travel time to health facilities providing blood type testing services in Ghana, especially in the UER. Knowledge of the distance and travel time to the nearest health facility for a blood type screening test is potentially essential to help the Government of Ghana implement POC testing services in rural health facilities located in geographical settings with poor access as recommended by the WHO and bring healthcare closer to where people live and work [21]. We, therefore, investigated the spatial accessibility to blood group and rhesus type POC testing during ANC in the UER of Ghana. 

## 2. Methods

### 2.1. Overview

This is a follow-up on our previously published cross-sectional study which investigated the availability and use of pregnancy-related POC tests in the UER utilising a hundred PHC clinics (health centres and CHPS compounds) randomly selected from a total of 356 clinics from all the districts in the region [20]. The sampling strategy used to select the 100 PHC clinics as well as the study area (UER) has been adequately described in the previously published survey [20]. The survey revealed that of the 100 PHC clinics, blood type screening test was available only in six clinics, and all six were health centres with some form of laboratory services [20]. To measure the spatial accessibility to health facilities where blood type screening testing is available in the regions, the cost-distance algorithm was applied using the ArcGIS desktop software. The flow diagram (Figure 1) illustrates the data, methods, and models used. All the attributes and spatial data used for this study were obtained in 2018.

We extracted the attribute data on all the 100 PHC clinics and health facilities where expectant mothers (who require blood group and rhesus status testing) were referred for blood type screening tests during ANC. These data which were obtained in text data format were loaded into the ArcGIS 10.5 software and transformed into a shapefile to allow for performing spatial analysis using ArcGIS. The topographic data gathered for this current study included roads, rivers, and the slope obtained as the digital elevation model (DEM). The topographical data were obtained for the whole of West Africa and the UER processed as a subset of Ghana. Road data were obtained to inform travel routes and, in view of this, types of roads were appended as attribute data to the spatial data to inform the potential speed users are likely to experience on each road type which may as well greatly inform travel distance and time. The DEM for the West Africa region was obtained to inform the slope of every location in the UER. This was necessary because identifying valleys and hills was key to determining which areas serve as barriers and are inaccessible to users. The DEM dataset was obtained from Adu Manu Kwame (AMK) Consultancy and compared with the data obtained from the University of Ghana Remote Sensing and Geographic Information Systems Laboratory for accuracy. We then reclassified the slope data into highlands (more than 200 m high) and flatlands (between 119 and 200 m high) as informed by the DEM data which showed that the highest point in the UER was about 470 m, and 119 m was the lowest point. 

### 2.2. Spatial Data of Health Facilities

To realise the objectives of this study, the geo-location data of all the 100 PHC clinics and health facilities such as public hospitals or clinics offering blood type testing services as well as private medical laboratories or hospital/clinics being utilised as referral points for blood type testing services were obtained from the Regional Health Directorate of the Ghana Health Service, UER. We also obtained the spatial data of all the 131 towns in the region using the global positioning system. To accurately map all the latitude and longitude to their relative location on the earth, the World Geodetic System 30 North coordinate system was applied to the entire dataset because Ghana falls within this zone. 

### 2.3. Geospatial Analysis

#### Developing a Model for Estimating the Travel Time

As a key aspect of this study, the model for determining the travel time was carefully developed taking into consideration all the datasets. Using the PyScripter integrated development environment and relying on the Python capabilities of ArcGIS 5.0, a model was developed to calculate the travel time and for data transformation. The cost distance model which calculates the shortest time to a source based on a cost dataset was used to determine distance. A motorised tricycle was identified as the commonly used public transport for travel within UER, hence, travel time was estimated via road and via paths and tracks using an assumed motorised tricycle speed of 20 km per hour. We recalibrated travel time per pixel (10 m × 10 m grid) for both roads and paths to enable estimation of travel time from PHC clinics where blood type testing services are not available to the nearest hospital, clinic, or medical laboratory for all the districts in the region. Although travel time can be influenced by many factors, we chose to estimate travel time via roads and paths because they are the commonly used routes in the region. Likewise, the motorised transport system was chosen because we found it was the most used public mode of transport for journeys within the region. The model and procedure used to approximate the travel time for this current study have been published in our previous studies focusing on geographical access to tuberculosis diagnostic services and comprehensive ANC POC diagnostic services [12,13]. Appendix A presents a detailed description of the procedure.

### 2.4. Buffer

We employed the geospatial analysis proximity tool (Buffer) to identify towns and PHC clinics without a blood group and rhesus type testing service located within a 10 km radius, and those located beyond 10 km to a health facility providing the service. We additionally estimated the proximity of all the 131 towns to the nearest health facility providing blood group and rhesus type testing service. Evidence shows that access to healthcare elsewhere more than 10 km away is associated with higher risks of poor health outcomes [22]. ‘Buffer’ in geographic information systems (GIS) refers to a boundary defined by specific units that surround a source or feature. For the purposes of this study, the point buffer was employed because the origin feature in this study which is health facilities providing blood type testing service is a point feature (vector dataset). 

To assess the accuracy, we created a set of random points from the ground truth data and compared that to the classified data in a confusion matrix using three geoprocessing tools: create accuracy assessment points, update accuracy assessment points, and compute confusion matrix.

### 2.5. Spatial Autocorrelation

We utilised the spatial autocorrelation tool in ArcMap 10.5 to determine the spatial distribution of the health facilities providing blood type testing services in the region. Spatial autocorrelation mirrors the first law of geography which states that “everything is related to everything else, but near things are more related than distant things” [23]. In spatial autocorrelation, the null hypothesis of the Moran’s Index statistic states that the feature being measured is distributed randomly, however, when the *p*-value obtained from running the spatial analyst tool proves to be statistically significant, then the null hypothesis can be rejected [24]. Guided by this, we considered a positive correlation to mean similar values clustered together while negative correlation is representative of different values clustered in a location, and zero means no correlation.

### 2.6. Ethics Approval

This study was approved by the Navrongo Health Research Centre Institutional Review Board/Ghana Health Service (approval number: NHRCIRB291) on 8th January 2018 and the University of KwaZulu-Natal Biomedical Research Ethics Committee (approval number: BE565/17) on 12th February 2018 

## 3. Results

### 3.1. Characteristics of the Health Facilities Providing Blood Group and Rhesus Type POC Testing

In all, blood group and rhesus type POC testing was available in 18 health facilities. Of the 18 health facilities, nine (50%) were hospitals and six (33.3%) health centres. The remaining three (16.7%) health facilities were private medical laboratories which were used as referral points by some of the PHC clinics without blood group and rhesus type POC testing service. Of the nine hospitals providing blood group and rhesus type POC testing services, the majority (77.8%) are owned and managed by the Ghana Health Service (GHS), 11.1% are owned and managed by the Christian Health Association of Ghana (CHAG), and 11.1% by private individuals. Similarly, 83.3% (5/6) of the sub-district health centres offering blood group and rhesus type POC testing are owned and managed by GHS, whilst one (16.7%) is owned by CHAG. All the 18 health facilities offering blood group and rhesus type POC testing services were distributed across nine out of the 13 districts in the region. Four (22.2%) each were in the Bongo district and Bolgatanga municipality; meanwhile, there was no health facility providing blood group and rhesus type POC testing services in Builsa South, Nabdam, Binduri, and Pusiga districts (Figure 2).

### 3.2. Spatial Distribution of Health Facilities Providing Blood Grouping and Rhesus Type Testing Services in the UER

To determine the spatial distribution of health facilities providing blood grouping and rhesus type testing services in the region, spatial autocorrelation analysis was conducted. The results showed a positive spatial autocorrelation (Moran Index = 0.29; z-score = 1.37; *p* = 0.17) suggesting that health facilities providing blood group and rhesus type testing services were randomly distributed spatially in the region (Figure 3). 

### 3.3. Spatial Accessibility to Blood Group and Rhesus Type Testing in the UER

We estimated the proximity (distance) of the 94 PHC clinics without blood group and rhesus type testing service from the previous cross-sectional study to the nearest health facility providing blood group and rhesus type testing services in the region. The results showed that 48 (51.1%) of the PHC clinics without blood group and rhesus type testing service were within 10 km reach to the nearest health facility offering the services. All the participating PHC clinics without blood group and rhesus type testing service from Bolgatanga Municipal and Bongo district were within 10 km range to the closest facility providing the service. However, none of the PHC clinics included in this study from the Builsa South district were within 10 km reach of any of the health facilities providing blood group and rhesus type testing service (Figure 4). 

Based on the PHC clinics included in this analysis, the mean (standard deviation (SD)) distance from a PHC clinic without blood group and rhesus type testing service to the closest facility providing the service in the UER was 12.6 ± 5.2 km. The longest mean distance was recorded in the Builsa South (32.2 ± 13.2 km) district, whilst the shortest (3.3 ± 1.4 km) was in the Bongo district. The results also show that the mean travel time from a PHC clinic without blood group and rhesus type testing service to the closest facility providing the service in the UER was 37.7 ± 15.4 min. Again, Builsa South district recorded the longest travel time (96 ± 39.4 min), whilst Bongo district recorded the shortest travel time (9.9 ± 4.1 min) to the nearest facility offering blood group and rhesus type testing service (Figure 5).

This study also estimated the proximity of all the 131 towns to their nearest health facilities providing blood group and rhesus type testing services. The results showed that 87 (66.4%) of the towns were within a 10 km radius (less than 30 min travel time) to a health facility providing blood group and rhesus type testing service. Twenty-five (19.1%) of the 131 towns were located between 10 and 15 km to a health facility offering blood group and rhesus type testing service, whilst 15 (11.5%) towns were located at more than 15–25 km reach. Meanwhile, 4 (3.1%) towns in Builsa South district were found to be located more than 25 km to the nearest health facility providing blood group and rhesus type testing service (Figure 6). Appendix A provides the distance/travel time categorisation of the towns and their names.

This study’s findings further showed that the mean distance and travel time ± SD from all locations in the 131 towns of the UER to a health facility providing blood group and rhesus type testing service in the region was 8.9 ± 4.1 km and the mean travel time was 17.8 ± 8.3 min. Builsa South and Bongo districts once again recorded the longest mean distance (25.6 ± 7.4 km) and the shortest (3.1 ± 1.9 km), respectively, to a health facility providing blood group and rhesus testing service in the region. Similarly, the mean travel time from all locations in the 131 towns of the UER to a health facility providing blood group and rhesus type testing services in the region was estimated at 17.8 ± 8.3 min with Builsa South district recording the longest travel time (51.3 ± 14.8 min) and the shortest time recorded in Bongo district (6.2 ± 3.9 min) (Figure 7).

## 4. Discussion

This study estimated the spatial accessibility (distance and travel time) to health facilities for blood group and rhesus type testing during ANC in the UER of Ghana. The results showed the about 51.1% of the PHC clinics without blood group and rhesus type testing and 66.4% of the towns in the region were within 10 km range to a facility providing the service. The mean distance to a health facility providing blood group and rhesus type testing service in the region was 8.9 ± 4.1 km, whilst the mean travel time was 17.8 ± 8.3 min using a motorised tricycle speed of 20 km/hour. The results further showed that the spatial autocorrelation of the health facilities providing blood group and rhesus type testing services were randomly distributed in the region.

We found the majority of the PHC clinics without blood group and rhesus type testing and the towns in the UER were within 10 km proximity to the nearest health facility providing the service. Although this finding is fairly good, at the same time, it depicts that almost half (48.9%) of the PHC clinics are without blood group and rhesus type testing facilities; moreover, about 33.6% of the towns are still geo-located beyond 10 km to a facility providing the services. This could also result in low utilisation of blood group and rhesus type testing services, overcrowding at the health facilities, and increased waiting time for test results. Evidence shows that people tend to limit the use of health services to facilities closer to them [25]. Ferguson and colleagues also demonstrated that having diagnostic technologies closer to populations streamlines critical care paths [26]. Hence, these results have revealed the need to equip the PHC clinics to enable them to provide POC testing for blood group and rhesus type tests in the region. Contrary to this current study finding, a previous study in the UER reported fewer PHC clinics were within 10 km reach to either a hospital or medical laboratory for a one-stop ANC POC diagnostic service [13].

We also found the mean distance to blood group and rhesus type POC testing in the region to be 8.9 ± 4.1 km and a mean travel time of 17.8 ± 8.3 min using a motorised tricycle speed of 20 km/hour. These findings suggest moderate spatial accessibility to blood group and rhesus type testing. It is possible that a substantial proportion of women of reproductive age live beyond 8.9 km from nearest health facility providing the service. The distance of 8.9 km and can take several hours for a pregnant woman to walk in the case where she is unable to afford the services of a motorised tricycle. Even when a pregnant woman can afford the services of a motorised tricycle but considering the bad state of the roads in the region, particularly in the rural areas, the speed of the tricycle may be less than the average 20 km/h used in our analysis. Therefore, the travel time to the nearest health facility may exceed the estimated travel time we found in some cases. In this case, it is possible that a significant number of referred pregnant women from PHC clinics may not go to the referral facilities for blood group and rhesus type testing during ANC and this potentially could become problematic when the need arises for urgent blood transfusion. Although we found no other study demonstrating evidence on spatial accessibility to health facilities for blood group and rhesus type testing, Gething and colleagues’ study on geographical access to care at birth in Ghana reported longer journey times to emergency obstetric and neonatal care [7]. Likewise, a study in South Africa by Tanser and colleagues also demonstrated poor physical accessibility (travel time of 170 min) to healthcare [10].

We further found the spatial autocorrelation findings also demonstrated the health facilities providing blood group and rhesus type testing services were randomly distributed in the region. This suggests the health facilities were neither clustered nor dispersed in the UER. However, our analysis showed district-wise variation in spatial accessibility to public hospitals and clinics offering blood group and rhesus type POC testing services in the region. Nonetheless, we have highlighted worst-served areas for improvement by the health authorities to address the disparities revealed, particularly, in the Builsa South district. Similarly, our previous studies in the UER which assessed the geographical access to a comprehensive ANC and tuberculosis POC testing also found geographic variations of the health facilities with moderate accessibility to diagnostic services [12,13].

Knowledge of blood group and rhesus type is of paramount importance to prevent transfusion-related complications particularly among pregnant mothers [27]. However, the findings revealed a low number of PHC facilities providing ABO and rhesus factor testing resulting in spatial variation of the service provision. Possible solutions to the geographical access and/or knowledge gap of pregnant mothers on ABO and rhesus factor type may include the implementation of POC testing for blood group and rhesus type in Ghana’s PHC clinics in accordance with the WHO recommendation. The WHO recommends a slide agglutination test using capillary whole blood or venous whole blood to determine A, B, and O groups and Rh type in resource-limited settings to facilitate the screening of patients, donors, safe blood transfusion, and improve access to healthcare and outcomes [21]. This will enable every pregnant woman and blood donors to be screened during ANC services for safe blood transfusion when needed urgently. The implementation of POC testing for blood group and rhesus type in Ghana’s PHC clinics may well reduce the distance and travel time to help improve access to healthcare generally and better maternal health outcomes in resource-limited settings. Additionally, the cost and risks of traveling long distances to access blood group and rhesus type testing services potentially will be reduced with a resultant increase in utilisation. Moreover, the implementation of POC testing of blood grouping and rhesus type for maternal healthcare may lead to a reduction of maternal deaths caused by haemorrhage, hence contributing to Ghana’s achievement of Sustainable Development Goal 3.1 (less than 70 maternal deaths per 100,000 live births) by 2030. According to WHO, countries such as Belarus, Bangladesh, Cambodia, Kazakhstan, Malawi, Morocco, Mongolia, Rwanda, Timor-Leste, and Zambia made substantial progress in reducing maternal mortality owing to the implementation of several interventions including their focus on improving the quality of care at the PHC level and universal health coverage [1].

Using geographical information systems (GIS) to inform the implementation of health services has been shown to be useful [28,29,30]. GIS effectively enables implementation of POC testing in health networks to streamline decision making at the POC [26]. Hence, GIS helps improve patient health outcomes, save time and money, and ensure that the health networks are sufficiently resourced to deliver needed health services to the population [26]. Despite these strengths, there are several limitations worth noting such as our inability to include non-spatial factors in the analysis. Non-spatial factors such as the income of patients [8], age, cultural practices, education, and others can also influence the utilisation of health services even if the service is very close to the individual [7]. Although the implementation of POC testing for blood group and rhesus type testing in PHC clinics has the potential to improve maternal health outcomes, this study did not assess the cost–benefit analysis and other challenges associated with the implementation. Nonetheless, knowledge of one’s blood group and rhesus type and safe blood transfusion timely could save many lives including those of pregnant women before, during, and after delivery. The travel time estimation provided by this study was dependent on only one mode of transportation using an assumed speed which might be inaccurate. Moreover, this study was limited to only one region in Ghana, and, therefore, the findings may not necessarily apply to the remaining fifteen regions in the country. We therefore recommend future research to focus on areas such as the non-spatial factors and cost–benefit analysis of implementing POC testing for blood group and rhesus type in PHC clinics. We also recommend a similar study in the other fifteen regions of Ghana. Notwithstanding these limitations, this study has provided evidence-based information useful for policy decision-making for targeted improvement of POC testing for blood group and rhesus type testing in the UER. Since this study is the first, it may possibly stimulate more research using GIS to evaluate access to blood group and rhesus type testing services in other similar settings for improving healthcare.

## 5. Conclusions

A blood group and rhesus type test are a prerequisite for both blood recipients and donors prior to transfusion to prevent or reduce blood-transfusion-related complications. The current mean distance and travel time to health facilities for blood group and rhesus type testing service in the UER is estimated at 8.9 ± 4.1 km, whilst the mean travel time was 17.8 ± 8.3 min using a motorised tricycle speed of 20 km/hour. This distance and travel time possibly may reduce substantially if blood group and rhesus type POC testing services are implemented in PHC clinics, particularly in those districts with poor spatial access evidenced by this study. We recommend the establishment and implementation of an essential diagnostic list including POC tests for blood group and rhesus type test in Ghana in line with the WHO recommendation to help address diagnostic challenges in resource-limited settings.

## Figures and Tables

**Figure 1 diagnostics-09-00175-f001:**
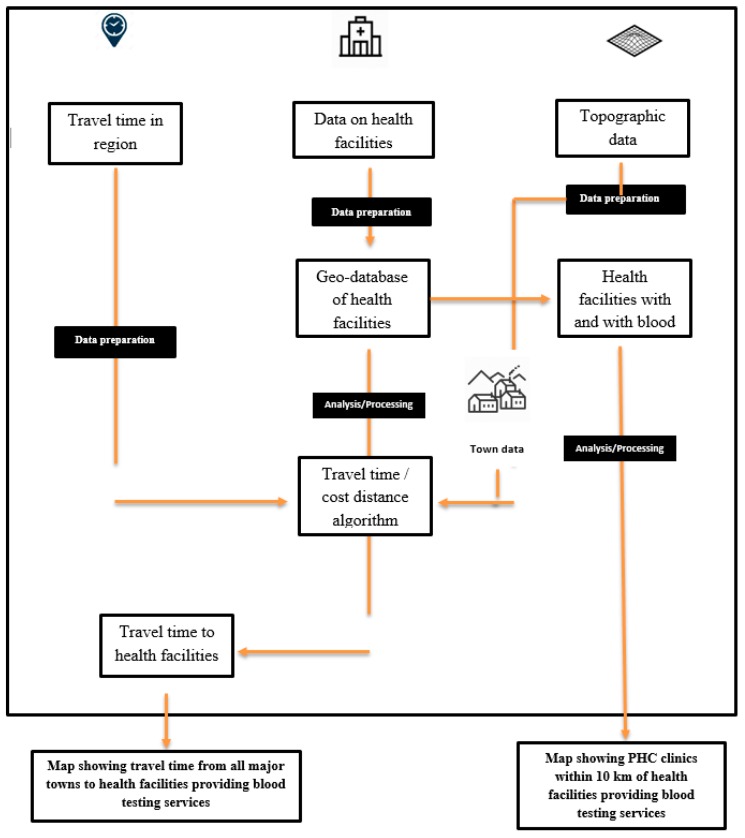
A flow diagram illustrating the data, methods, and models used. PHC—primary healthcare.

**Figure 2 diagnostics-09-00175-f002:**
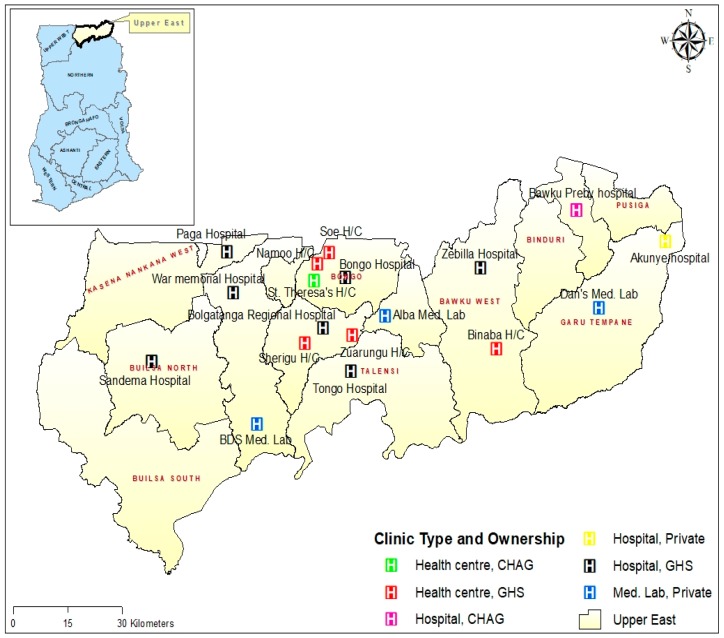
Map showing the geographical location, facility type, and ownership of health facilities providing blood type testing services in the Upper East Region (UER). CHAG—Christian Health Association of Ghana; GHS—Ghana Health Services.

**Figure 3 diagnostics-09-00175-f003:**
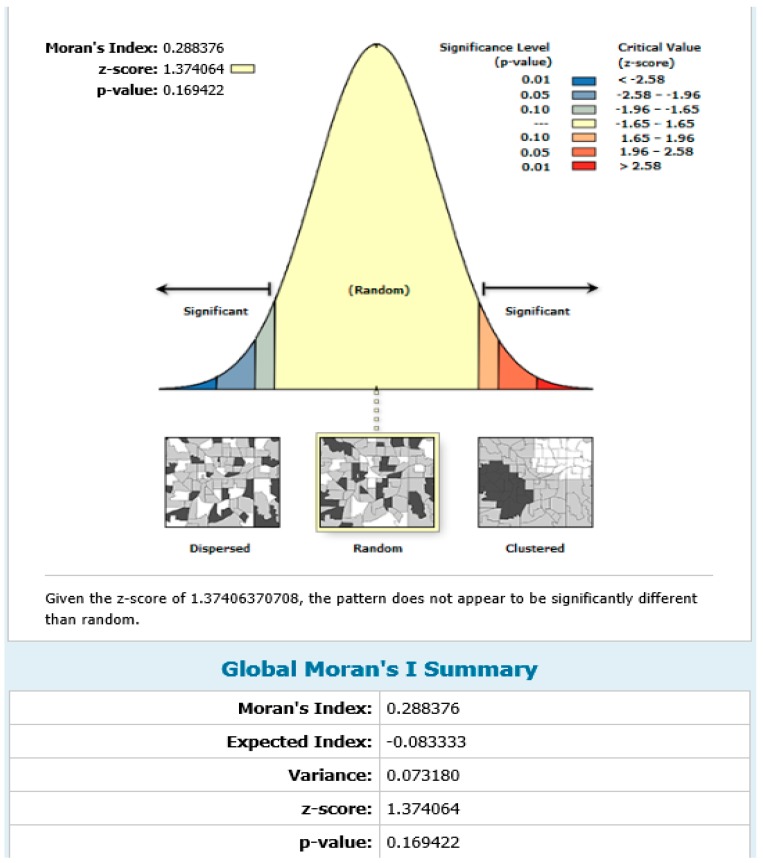
Spatial autocorrelation map showing the spatial distribution of health facilities providing blood group and rhesus type testing in the UER.

**Figure 4 diagnostics-09-00175-f004:**
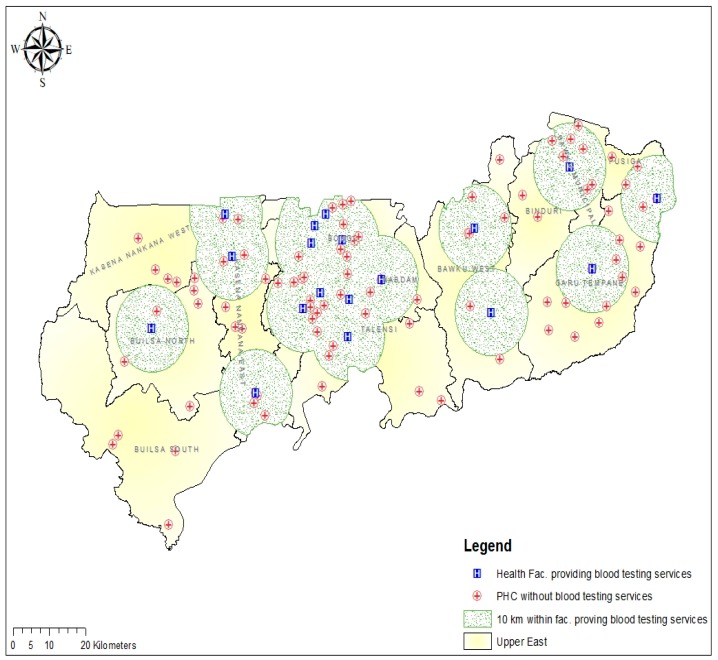
Map showing the distribution of the PHC clinics and distance within 10 km from the nearest health facility providing blood group and rhesus type testing services in the UER.

**Figure 5 diagnostics-09-00175-f005:**
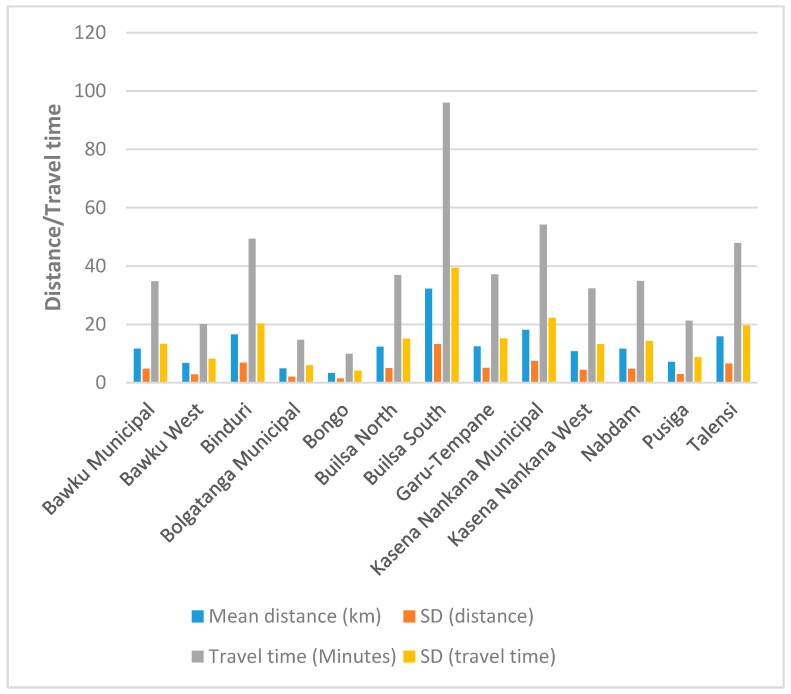
A bar chart depicting the mean distance and travel time from PHC clinics to the nearest health facility providing blood group and rhesus type testing services per district in the UER.

**Figure 6 diagnostics-09-00175-f006:**
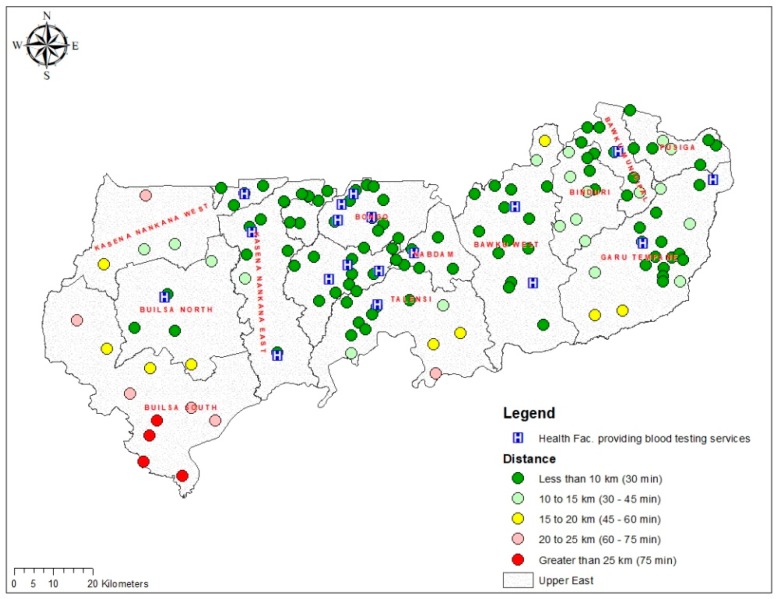
Map showing the distribution of distance (km) and travel time (minutes) from towns to health facilities providing blood group and rhesus type testing services in the UER.

**Figure 7 diagnostics-09-00175-f007:**
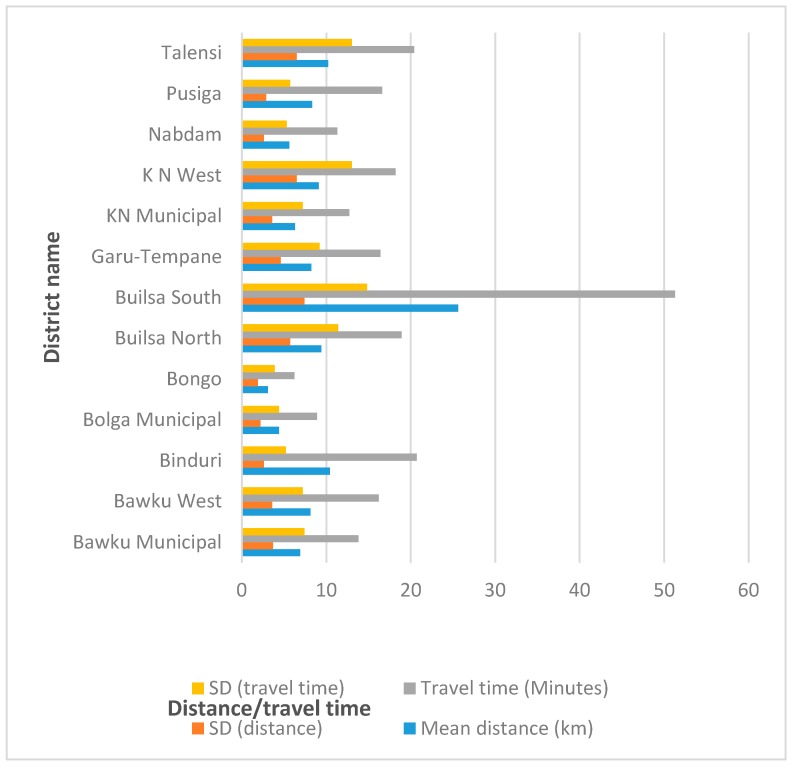
A bar chart depicting the mean distance and travel time from all 131 towns in the UER to a health facility providing blood group and rhesus type testing services per district in the region.

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
