# Peer review of "Estimating the Spatial Accessibility to Blood Group and Rhesus Type Point-of-Care Testing for Maternal Healthcare in Ghana"

_diagnostics, 2019, doi:10.3390/diagnostics9040175_

Round 1

Reviewer 1 Report

This study estimated the spatial accessibility to health facilities providing blood group and type testing services as an integral part of maternal health care in the Upper East Region (UER), Ghana. The distance and travel time to the nearest health facility for a blood type screening test were estimated. In my opinion, the work has some weakness which should be addressed before consideration of publication.

First of all, the abstract is too long. Please exclude unimportant information.

What does health inequality mean and how was it defined in this study?

If the data is available, please add the current information/statistics showing the number of people taking blood type screening test in each health service.

Besides of spatial accessibility which is important to promote people’s decision to take blood type screening services, I think there are other important factors which could affect people’s use of blood type screening services. Those are such as income, status, education level etc. This should be reviewed and added in the literature review part which was put in the introductory part. Though, it was mentioned in the limitation of the study, I think they should be mentioned before, and concrete reason to not include those factors, and to emphasize only some factors should be provided.

Similarly, there are diverse factors affecting travel time. They are not just only mode of transportation, distance, types of roads, and topology, but also, land use, building use, traffic condition, and population density (Which is very important). I am curious why they were not included in the analysis. Please do more literature reviews, and add this issue in the in section “2.3.1. Developing a model for estimating the travel time”.  In the last paragraph of this section, you may summarize by providing the reason why only mode of transportation, distance, types of roads, and topology.

Please add the year of Geographical data used for this analysis?

Page 5 “2.6. Ethics approval and consent to Participate” Who are participants of this study?

Do you check the accuracy of the assessment carried out through Spatial Analysis (GIS)? The work really emphasized on spatial analysis based on GIS, therefore, the accuracy of assessment should be carried out.

The research would be more interesting if the correlation between the numbers of people taking the services and estimated distance and travel time to the nearest health facility for a blood type screening test is analyzed and shown. This could confirm that spatial accessibility to health facilities have an effect on people’s use of blood type screening services.

In the discussion part, the results showed that majority of the PHC clinics without blood group and rhesus type testing and the towns in the region were within 10 km range to a facility providing the service. Please discuss whether it is good or not.

The mean distance to a health facility providing blood group and rhesus type testing service in the region was estimated at 8.9 km ± 4.1 whilst the mean travel time was 17.8 minutes ± 8.3 using a motorised tricycle speed of 20 km/hour. Please discuss whether it is good or not.

The spatial autocorrelation findings also demonstrated the health facilities providing blood group and rhesus type testing services were randomly distributed in the region. However, the results suggest inequalities in the distribution of and spatial accessibility to public hospitals and clinics offering blood group and rhesus type testing services across the districts in the region. Please explain potential adverse consequences of this situation.

Please make recommendations for policy development in order to tackle with this situation.

Author Response

We are most grateful to you for finding time out of your busy schedule to review this manuscript. Your comments and suggestions were very useful and have helped improve this revised submission significantly. Please have attached our responses for your consideration.

Responses to Reviewer #1 Comments and Suggestions

General comment

This study estimated the spatial accessibility to health facilities providing blood group and type testing services as an integral part of maternal health care in the Upper East Region (UER), Ghana. The distance and travel time to the nearest health facility for a blood type screening test were estimated. In my opinion, the work has some weakness which should be addressed before consideration of publication.

General response

On behave of the authors, I wish to express my profound gratitude to you for finding time out of your busy schedule to review this manuscript. We sincerely appreciate your commendations, valuable comments, and suggestions to improve the quality of this manuscript. We have accordingly addressed all your comments as indicated in the revised submission for your consideration.

Comment 1

First of all, the abstract is too long. Please exclude unimportant information.

Response

We have edited the abstract based on your suggestion.

Comment 2

What does health inequality mean and how was it defined in this study?

Response

Thank we meant to say geographic variations/disparities. We revised it throughout the manuscript.

Comment 3

If the data is available, please add the current information/statistics showing the number of people taking blood type screening test in each health service.

Response

We have included this under the revised introduction section (Page 4, LN 108-110).

Comment 4

Besides of spatial accessibility which is important to promote people’s decision to take blood type screening services, I think there are other important factors which could affect people’s use of blood type screening services. Those are such as income, status, education level etc. This should be reviewed and added in the literature review part which was put in the introductory part. Though, it was mentioned in the limitation of the study, I think they should be mentioned before, and concrete reason to not include those factors, and to emphasize only some factors should be provided.

Response

We sincerely appreciate your suggestion. We have revised the background to include this. Please Pages  3 and 4, LN 73-111

Comment 5

Similarly, there are diverse factors affecting travel time. They are not just only mode of transportation, distance, types of roads, and topology, but also, land use, building use, traffic condition, and population density (Which is very important). I am curious why they were not included in the analysis. Please do more literature reviews, and add this issue in the in section “2.3.1. Developing a model for estimating the travel time”.  In the last paragraph of this section, you may summarize by providing the reason why only mode of transportation, distance, types of roads, and topology.

Response

We are most grateful. We have partly addressed this in the revised background. Please see pages 3 and 4, LN 73-111 and as well as at the methods section Page 6, LN 175-178

Comment 6

Please add the year of Geographical data used for this analysis?

Response

Please we have included this (Page 5, LN 134-135)

Comment 7

Page 5 “2.6. Ethics approval and consent to Participate” Who are participants of this study?

Response

Thank you. This was inserted in error. This was done in our initial survey. We have deleted it.

Comment 8

Do you check the accuracy of the assessment carried out through Spatial Analysis (GIS)? The work really emphasized on spatial analysis based on GIS, therefore, the accuracy of assessment should be carried out.

Response

We conducted an accuracy assessment using three geoprocessing tools: create accuracy assessment points, update accuracy assessment points, and compute confusion matrix. Please see page 7, LN 194-196.

Comment 9

The research would be more interesting if the correlation between the numbers of people taking the services and estimated distance and travel time to the nearest health facility for a blood type screening test is analyzed and shown. This could confirm that spatial accessibility to health facilities have an effect on people’s use of blood type screening services.

Response

We sincerely agree with you. However, the antenatal attendance data we have is for all the women who attended these clinics for various antenatal services including blood group and rhesus type testing. So do not have sufficient data to conduct correlation for this study. Nonetheless, we have noted this for subsequent studies. Thanks so much for the idea.

Comment 10

In the discussion part, the results showed that majority of the PHC clinics without blood group and rhesus type testing and the towns in the region were within 10 km range to a facility providing the service. Please discuss whether it is good or not.

Response

Thank you. We have addressed this concern. Please see the revised discussion section for confirmation (Pages 17 and 18, LN 335-375

Comment 11

The mean distance to a health facility providing blood group and rhesus type testing service in the region was estimated at 8.9 km ± 4.1 whilst the mean travel time was 17.8 minutes ± 8.3 using a motorised tricycle speed of 20 km/hour. Please discuss whether it is good or not.

Response

Thank you. We have addressed this concern. Please see the revised discussion section for confirmation (Pages 17 and 18, LN 335-375

Comment 12

The spatial autocorrelation findings also demonstrated the health facilities providing blood group and rhesus type testing services were randomly distributed in the region. However, the results suggest inequalities in the distribution of and spatial accessibility to public hospitals and clinics offering blood group and rhesus type testing services across the districts in the region. Please explain potential adverse consequences of this situation.

Response

Thank you. We have addressed this concern. Please see the revised discussion section for confirmation (Pages 17 and 18, LN 335-375

Comment 12

Please make recommendations for policy development in order to tackle with this situation

Response

Thank you. We have addressed this concern. Please see highlighted under the conclusion section for confirmation (Pages 20, LN 429-432).

Reviewer 2 Report

This is an interesting paper that investigates the inequalities in spatial access to blood and RH test in Ghana. Thanks for the work. Here are a few comments that can help improve the manuscript. 

A big portion of the introduction was invested on discussing the maternal death issues, plus there is almost no information on the causes of unequal access to these services and what are the particularities about this setting. I trust the text needs to be revised as authors are missing the chance to inform readers regarding the relevant literature. 

The methods and results look good. 

Since the study is about assessing spatial accessibility, this naturally comes with the question about quality of roads and transportation facilities. Authors should take this factor into consideration in interpreting the findings.

How big is the problem and associated healthcare challenges that arise from not testing  blood group and unsafe blood transfusion for pregnant women in Ghana? Are there any countermeasures in place?

The discussion can be made more reader friendly by starting with a brief description on the main findings, and ending by discussing the main implications. Currently the text is mostly a review of other papers and fails to critically discuss the current findings. 

I hope this helps. 

Author Response

We are most grateful to you for finding time out of your busy schedule to review this manuscript and for providing us with useful comments to improve this revised manuscript significantly. We have attached our responses for your consideration. Thank you once again.

Responses to Reviewer #2 comments and suggestions

General comment

This is an interesting paper that investigates the inequalities in spatial access to blood and RH test in Ghana.

General response

On behave of the authors, I wish to express my profound gratitude to you for finding time out of your busy schedule to review this manuscript. We sincerely appreciate your commendations, valuable comments, and suggestions to improve the quality of this manuscript. We have accordingly addressed all your comments as indicated in the revised submission for your consideration.

Comment 1

Thanks for the work. Here are a few comments that can help improve the manuscript. A big portion of the introduction was invested on discussing the maternal death issues, plus there is almost no information on the causes of unequal access to these services and what are the particularities about this setting. I trust the text needs to be revised as authors are missing the chance to inform readers regarding the relevant literature.

Response

Thank you. Please we have revised the background based on your recommendations and the first reviewer. Please Pages 3 and 4, LN 73-111.

Comment 2

The methods and results look good.

Response

We sincerely appreciate your commendation.

Comment 3

Since the study is about assessing spatial accessibility, this naturally comes with the question about quality of roads and transportation facilities. Authors should take this factor into consideration in interpreting the findings.

Response

Thank you. We have revised the discussion section based on your comment as well as the first reviewer. This is highlighted in the manuscript pages 17 and 18, LN  335-375.

Comment 4

How big is the problem and associated healthcare challenges that arise from not testing blood group and unsafe blood transfusion for pregnant women in Ghana? Are there any countermeasures in place?

Response

Thank you. The problem and associated healthcare challenges that arise from not testing blood group and unsafe blood transfusion for pregnant women in Ghana have not been documented. However, haemorrhage especially post-partum haemorrhage is a major course of maternal death in Ghana. So every woman during antenatal is required to undergo blood group and rhesus type testing among other screening tests in preparation for transfusion if the need arises. So it is imperative for all health facilities irrespective of their location since they render maternal health services. We have captured all of this under the background. However, your comment has raised a useful research question worth considering in future.

Comments 5

The discussion can be made more reader friendly by starting with a brief description on the main findings and ending by discussing the main implications. Currently the text is mostly a review of other papers and fails to critically discuss the current findings.

Response

Thank you for your observation. As earlier stated, we have revised the discussion section. This is highlighted in the manuscript pages 17 and 18, LN  335-375.

Comment 6

I hope this helps.

Response

Your comments and suggestions have greatly helped to improve this manuscript. We are most grateful.

Round 2

Reviewer 1 Report

The paper is substantially improved, and I think it can be accepted for publication in its current form.